# The Role of Lung Colonization in Connective Tissue Disease-Associated Interstitial Lung Disease

**DOI:** 10.3390/microorganisms9050932

**Published:** 2021-04-27

**Authors:** Alberto Ricci, Alessandra Pagliuca, Morgana Vermi, Dario Pizzirusso, Marta Innammorato, Raffaele Sglavo, Francesco Scarso, Simonetta Salemi, Bruno Laganà, Roberta Di Rosa, Michela D’Ascanio

**Affiliations:** 1U.O.C. Pneumologia, Dipartimento di Medicina Clinica e Molecolare, Sapienza Università di Roma, A.O.U. Sant’Andrea, 00189 Rome, Italy; a.pagliuca90@gmail.com (A.P.); morganavermi@virgilio.it (M.V.); dario.pizzirusso@gmail.com (D.P.); marta.innammorato@hotmail.it (M.I.); raffaele.sglavo@gmail.com (R.S.); francesco.scarso@uniroma1.it (F.S.); dascaniomichela87@gmail.com (M.D.); 2U.O.C. Medicina Interna, Dipartimento di Medicina Clinica e Molecolare, Sapienza Università di Roma, A.O.U. Sant’Andrea, 00189 Rome, Italy; simonetta.salemi@uniroma1.it (S.S.); bruno.lagana@uniroma1.it (B.L.); roberta.dirosa@uniroma1.it (R.D.R.)

**Keywords:** interstitial lung diseases, connective tissue diseases, autoimmune disease, microbiota, pulmonary fibrosis, lung function tests, high-resolution computed tomography

## Abstract

Connective tissue diseases (CTDs) may frequently manifest with interstitial lung disease (ILD), which may severely impair quality and expectation of life. CTD-ILD generally has a chronic clinical course, with possible acute exacerbations. Although several lines of evidence indicate a relevant role of infections in the acute exacerbations of CTD-ILD, little information is available regarding the prevalence of infections in chronic CTD-ILD and their possible role in the clinical course. The aim of the present retrospective study was the identification of lung microbial colonization in broncho-alveolar lavage from patients affected by stable CTD-ILD with radiologically defined lung involvement. We demonstrated that 22.7% of patients with CTD-ILD display microbial colonization by *Pseudomonas aeruginosa*, *Haemophilus influenzae,* and non-tuberculous mycobacteria. Moreover, these patients display a major radiologic lung involvement, with higher impairment in lung function tests confirmed in a multivariate logistic regression analysis. Overall, the present study provides new information on lung colonization during CTD-ILD and its possible relationship with lung disease progression and severity.

## 1. Introduction

Interstitial lung disease (ILD) is a common and severe manifestation of systemic autoimmune connective tissue diseases (CTDs) such as rheumatoid arthritis (RA), systemic sclerosis (SSc), systemic lupus erythematosus (SLE), polymyositis/dermatomyositis (PM/DM), Sjögren syndrome (SS), mixed connective tissue disease (MCTD), and anti-neutrophil cytoplasm antibodies (ANCA)-vasculitis [1,2]. It may appear during an already established CTD or even as the first and sometimes unique CTD manifestation; in this case, the diagnosis is more difficult and may be delayed. ILD generally has a severe clinical course, which heavily influences the patient’s quality of life and mortality. It may appear as a complication of the treatment generally administered to CTD, in particular methotrexate, but even biologics and leflunomide may be implicated [1,3]. Some patients with ILD and auto-antibodies without CTD criteria may have a lung predominant feature as an expression of a CTD. Therefore, since 2013 the American Thoracic Society/European Respiratory Society (ATS/ERS) consensus on interstitial lung disease has clearly identified this clinical/radiological condition [4]. CTD-ILD may generally present with the histopathologic pattern of non-specific interstitial pneumonia (NSIP), but it may even present as usual interstitial pneumonia (UIP), especially in RA, in which it is the prevalent pattern [1]. UIP seems to have a more aggressive clinical course compared to NSIP. ILD generally has a chronic course, but it may even show up acutely; in this case, it may be either an exacerbation of a chronic CTD-ILD or a new presentation. The recommended therapeutic approach in the case of acute ILD considers the use of broad-spectrum antibiotics, thus underlining the role that bacterial infections may play in the ILD exacerbations. Although the relevant role of infections in the natural history of CTD-ILD is well recognized, little is known regarding their prevalence and their capacity to influence the severity of CTD-ILD.

The aim of the present investigation was the retrospective study of the lung colonization prevalence in patients affected by CTD-ILD. Moreover, the relationship between colonization and radiologic and functional impairment was also analyzed.

## 2. Materials and Methods

### 2.1. Patients and Sample Collection

Patients with a history of CTD-ILD were retrospectively studied (from 2010 to 2019). The diagnosis was based on clinical, serological, and radiological criteria [5]. Smoking status and drug use were also investigated. Patients with CTD-ILD presented exertional dyspnea and a nonproductive cough. Extra-pulmonary symptoms (arthralgia, myalgia, fatigue), the presence of skin findings (skin thickening and telangiectasia), Raynaud’s phenomenon, discoid or butterfly rashes, and heliotrope rash were reported [6]. Pulmonary auscultation revealed crackles at the pulmonary basis. High-resolution computed tomography (HRCT) scanning was performed to evaluate lung involvement. The diagnosis of CTD-ILD was reached after a multi-disciplinary consensus [7]. A specific computed assisted quantification of the radiological finding is detailed below.

The hospital ethics committee approved the study (11 October 2010–504/10). All the subjects signed informed consent.

### 2.2. Broncho-Alveolar Lavage Procedure

Flexible bronchoscopy (FBS) was used to harvest broncho-alveolar lavage (BAL) fluid using standard procedures. Briefly, after wedging the bronchoscope into the sub-segmental bronchi of the involved lung, BAL was conducted instilling three or four 50 mL aliquots of sterile 0.9% warm saline solution through the working channel of the bronchoscope. The first aliquot, predominantly representing airway cells and secretions, was discarded. The retrieved fluid (aliquots 2 and 3), which was more representative of the distal airway spaces, was sent to the laboratory of microbiology for quantitative bacterial culture. BAL fluid was cultured for aerobic and anaerobic bacteria, fungi (*Aspergillus* and *Candida* species), and mycobacteria. A bacterial concentration of 1 × 10^3^ colony forming units (CFU)/mL or higher in BAL fluid allows us to identify the bacteria present. We considered a quantitative culture positive when bacteria were present at a concentration of 1 × 10^4^ CFU/mL or higher. In fact, considering that BAL fluid is the expression of a dilution of lung secretions of 1:10 to 1:100 [8] and that the growth of pathogen bacteria in lung secretions is considered significant at between 1 × 10^5^ and 1 × 10^6^ CFU/mL, the cut-off of 1 × 10^4^ has been chosen as significant [9]. True pathogens were classified as aerobic Gram-positive bacteria, aerobic-Gram negative bacteria, and atypical pathogens. Mycobacterial species were detected by the identification of the acid-fast bacilli using the Ziehl–Neelsen technique, and cultures on special media (Lowenstein–Jensen is required). A more sensible molecular analysis based on the polymerase chain reaction (PCR) was used to identify tuberculous and non-tuberculous mycobacteria (NTM) organisms.

### 2.3. Lung Function Tests

All the patients performed pulmonary function tests according to a well-defined protocol and the ATS/ERS recommendations [10]. We used the plethysmography method to evaluate the static and dynamic pulmonary volumes (particularly forced vital capacity, FVC), while the lung diffusion capacity of carbon monoxide (DLCO), evaluated using the single breath-hold method, was used to verify the status of the blood–air barrier.

### 2.4. Quantification of Lung CT Lesions

The results of the HRCT scans were quantified using a modified semi-quantitative scoring system (Lung Quantitative Software—Siemens). Using this software, we quantitatively assessed pulmonary involvement according to lung lesion distribution and size. A total of 20 lung segments in both the left and right lungs were analyzed. The lesion size score was based on more than 50% occupation by the lesion of the lung segment (Total Severity Score (TSS)). The score was 1 for ≥50% involvement and zero for <50%, with a maximum total score of 20. The higher the value, the more severe the inflammatory load. 

### 2.5. Statistical Analysis

The data were collected in a dataset assembled using Microsoft Office package 2019 (Excel 2019 v. 16.3). All statistical analyses were conducted using the commercial statistical software SPSS and R version 3.6.1. All graphs were created using the software ‘R’ version 3.6.1 for Windows. Descriptive statistical analysis was performed on raw data where applicable. Results were expressed as means ± standard deviation (SD). Analyses were performed using Pearson’s χ^2^ test for categorical variables and Student’s t test for continuous variables. A two-tailed *p* value of 0.05 or less was used as a criterion to indicate statistical significance. Data were statistically analyzed using the GraphPad software (GraphPad Software, San Diego, CA, USA). Univariable odds ratios (ORs) and associations of continuous variables with BAL findings were calculated using logistic regression analysis. A multivariate logistic regression analysis was carried out referring to lung infections as the dependent variable and adjusting for demographic characteristics and lung function tests as covariates. Significance was calculated with the Wald Test. 

To avoid the unmeasured confounders influencing the results and to explain the reported associations, we carried out a sensitivity analysis to address the effect of the confounding factors. Therefore, the E-value was used as an approach in the sensitivity analyses to quantify unmeasured confounding in this observational study [11,12]. The E-value provides a measure related to the evidence of causality.

## 3. Results

### Patients

Patients affected by CTD, referred to Sant’Andrea University Hospital from 2010 to 2019, were studied. Among 150 CTD patients, 44 had a diagnosis of clear interstitial lung involvement. We retrospectively evaluated data obtained from these patients affected with CTD-ILD. The socio-demographic, lifestyle, and clinical characteristics are reported in Table 1. Smoking status and drug use were investigated. The CTDs were represented by SSc (*n* = 18), RA (*n* = 10), overlap syndrome (*n* = 6), SS (*n* = 1), ANCA-vasculitis (*n* = 3), and MCTD (*n* = 6). The mean age at diagnosis was 52.7 ± 16 years old, with a female predominance (F/M = 34/10). CTD was diagnosed before lung involvement in nine cases, whereas it was diagnosed after ILD in eight patients. Diagnosis was contemporaneous in 27 patients.

All these patients underwent fiber optic bronchoscopy for diagnostic purposes, without lung acute exacerbation at the moment of the procedure. BAL was performed for the total and differential cell count and microbiology. HRCT was performed in all the cases. Representative CT scans are reported in Figure 1. The pulmonary function test findings detected in the CTD-ILD patients demonstrated a restrictive pattern in many cases (39/44; 88.6%), with a reduced DLCO (Figure 2). However, in the presence of other thoracic compartment involvement, such as in the airways, pulmonary vascular tree, or chest wall, different functional patterns may occur. Sometimes, a disproportionate reduction in DLCO may be observed in CTD patients with coexistent emphysema, while a significant reduction in lung volumes with a relatively preserved DLCO was documented for extrapulmonary restriction (chest wall skin thickening, respiratory muscle weakness). 

Although each CTD had a predilection for a specific radiological pattern, a radiographic NSIP pattern was commonly found (*n* = 30; 68%).

Ground glass opacities and intra- and inter-lobular reticular opacities with basal and sub-pleural distribution are present at a higher degree. Reticulation, traction bronchiectasis, and honeycombing reflecting fibrotic changes were less frequently observed (*n* = 11; 25%) (Figure 1). Rarely was an indetermined pattern described (*n* = 3; 7%). 

No fungi were identified, whereas in 10/44 (22.7%) patients studied by FBS and BAL the presence of bacteria, in particular *Pseudomonas aeruginosa* (*n* = 4; 9%), *Haemophilus influenzae* (*n* = 3; 6.8%), and NTM organisms (*n* = 3; 6.8%), was detected (Table 2). 

The 10 patients with lung colonization included more males older than the 34 CTD-ILD patients without lung colonization, as observed in the univariate analysis (Table 1). Current or former smoking habit did not represent a risk factor for lung colonization, as well as having SSc or RA (Table 1). Unfortunately, incomplete information was found on the comorbidities and therapeutic regimens, which did not allow us to evaluate them as possible risk factors. Bacterial colonization was associated with more compromised HRCT lung involvement (Table 1 and Figure 1). The representative HRCT scan images and histograms represent data obtained from radiologic and functional assessment in CTD-ILD patients, respectively, in the absence (Group 1) or presence (Group 2) of colonization in BAL fluid. Note the significantly (*p* = 0.0398) higher radiological involvement expressed in terms of CT lung involvement and TSS in Group 2 than in Group 1 patients. Lung function assessments in these patients demonstrated a more compromised involvement, expressed in terms of FVC and DLCO (Table 1 and Figure 2), in comparison to patients who did not show bacteria colonization. A clear statistically significant lung functional test impairment in Group 2, expressed in terms of FVC and DLCO/alveolar volume ratio (DLCO/VA) (*p* = 0.01142 and *p* = 0.00841, respectively), was observed (Table 1 and Figure 2). 

The univariate logistic regression analysis was only carried out in patients with SSc and RA. In fact, patients with SS, Overlap Syndrome, ANCA-vasculitides, and MCTD (*n* = 1, 6, 3, 6 patients, respectively) were excluded for the low number of cases. The results of the analysis associated male gender (with an OR of 11.25) and respiratory dysfunction (FVC and DLCO) with risk of developing bacterial colonization contrarily to age, smoking habit, and steroid use. 

However, in a multivariate logistic regression analysis, DLCO and FVC were the only covariates still maintaining statistical significance (the results obtained using DLCO and FVC values are shown in Table 3 and Table 4). We did not insert in the multivariate logistic regression analysis DLCO and FVC together. Their values were considered separately for their strict association, as demonstrated by the linear regression analysis performed. 

A sensitivity analysis to address the effect of confounding factors in the reported associations was performed. For this reason, we calculated the E-value. The OR for a DLCO increase of one percentage point was 0.8366 (95% C.I. 0.72/0.925), with an estimated E-value of 1.41 (with a 1.21 E-value for the upper limit). The same results were obtained using the FVC values. We found an OR of 0.607 (95% C.I. 0.37/0.795) with an estimated E-value of 1.86 (with 1.49 E-value for the upper limit).

## 4. Discussion

ILD observed in the course of autoimmune diseases is a relatively newly defined entity. In these patients, the presence of microorganisms, at the pulmonary level, raises a series of questions with respect to the immunosuppressive therapeutic regimens used. On the other hand, little is still known about the mechanisms that determine lung interstitial damage, and the role of lung infections and bacterial colonization that can interfere with pulmonary function and prognosis [13,14,15].

Our data show that almost a quarter of patients affected with CTD-ILD had bacterial colonization in the lower respiratory tract associated with altered lung function. However, no demographic or lifestyle characteristics appeared to be associated with an increased risk of developing lung colonization, as inferred by the multivariate logistic regression analysis. In an analysis of risk factors for NTM in RA also, smoking has not been identified as a risk factor [16]. Unfortunately, data on comorbidities and drug use did not allow to calculate any possible association regarding risk factors and clinical severity of the colonized ILD. These results may be affected by the relatively low number of patients on immunosuppressive treatment, probably reflecting the minority of patients who had been diagnosed with CTD before the onset of respiratory symptoms. Moreover, the strength of unmeasured confounders was clarified by sensitivity analysis in order to be assured of the reliability of the reported associations. The E-value, evaluating the robustness of the highlighted effect, demonstrated the strength of our result. 

However, bacterial infection was recently identified as a risk factor for worst outcome in patients with idiopathic inflammatory myopathy-associated ILD, whereas the treatment with steroid and disease-modifying anti-rheumatic drugs resulted in being protective in relation to the short-term mortality [17]. Conversely, oral steroids were found to represent a significant risk factor for NTM disease in RA [14,18], thus underlining that few certainties exist and the relatively few studies on the topic are scarcely comparable and unable to provide definitive conclusions. 

Considering that all the patients described here underwent bronchoscopy and BAL in the context of a diagnostic purpose, but not for an acute exacerbation of chronic lung disease, we indirectly assumed that the observed infections may be the expression of a lung colonization. The colonization is a condition characterized by high concentration of microorganisms in the absence of clinical symptoms, more than a true infection from pathogens [19]. However, this distinction is mainly based on the clinical picture, but it is not easy to have a confirmation, depending on a series of variables, including the patient immune response [19,20], competition at the site by other organisms, and use of antimicrobials [19]. Moreover, it cannot be excluded that a non-negligible percentage of patients will develop clinical disease following colonization. Colonization can persist for months or years and chronic pulmonary diseases are often complicated with the bacterial colonization of the lower airways. This condition necessarily influences the therapeutic approach and prognosis [21,22]. Unfortunately, a lack of information on the follow-up prevents any conclusions from being drawn on this aspect. 

Recently, the growing development of culture-independent molecular analysis by the 16S ribosomal RNA gene [23,24,25] has allowed us to study the bacterial communities in several human niches, such as the gastrointestinal tract, oropharynx, vagina, respiratory tract, and skin [26,27,28], with unprecedented precision. This has allowed us to define the richness of bacterial communities in healthy subjects even in districts, such as the lung, which were considered sterile until a few years ago [29]. The study of the lung microbiota has been slower and less developed than the study of the microbiota of other more easily accessible body districts. In fact, the BAL analysis technical approach to reach distal airways, which is largely used, even in the current study, has been criticized for the fear of contamination by the upper airways’ microbial flora. However, in addition to discarding the first retrieved fluid aliquot, further evidence against possible contamination is the observation that entering the bronchoscope through the nose or mouth, which are markedly different in their respective microbiota [24,30], is totally uninfluential on the observed BAL microbiota [31]. In different lung pathologies, a microbiota associated with a specific pathology has been described [32]. Furthermore, the acute exacerbation of chronic lung diseases has been interpreted as the loss of balance between harmless/protective and pathogenic bacteria, a condition defined as dysbiosis, related to local environmental conditions [32]. A recently proposed model based on the results from the culture-independent molecular analysis of healthy and diseased lung microbiota, considers the terms “infection”, “colonization”, and “health” as outdated for understanding the meaning of the presence and origin of lung bacteria, whereas a better comprehension may come from putting active infection and healthy lung microbiota as the extremes of a unique line, in which the major or minor bacterial involvement is only the expression of different degrees of dysbiosis [29]. Moreover, the existence of a gut–lung axis has been recently identified [33]. The selective growth in a cultural analysis of *P. aeruginosa* and *H. influenzae* may therefore be considered the expression of a lung dysbiosis, probably connected to gut dysbiosis. If this interpretation better describes the biology of the phenomenon, the potential relevance of an innovative preventive intervention based on probiotic and prebiotic administration may be hypothesized. In fact, probiotic and prebiotic have been proven to be able to restore the normal microbiota [32,33], whereas it is well known that antibiotics are able to induce and maintain dysbiosis. 

This study showed that lung colonization in patients with CTD-ILD may be associated with more severe ILD. It is difficult to clarify if this condition is an epiphenomenon or a specific local condition observed during a more severe stage of CTD-ILD. Lung diseases may represent a favorable habitat for different microbial flora able to dominate the normal lung microbiome. At the present time, it is difficult to interpret whether this phenomenon is due to the reduced competitiveness of the commensal flora during disease status, the persistence of aberrant ecological niches in chronic diseases [33], or the structural re-arrangement of the lung parenchyma and the respiratory epithelium [34]. Several lines of evidence indicate an increased overall infection risk in patients with RA-ILD [35]. Although in our group of patients we have not found any cases of aspergillosis, some data report the role of this pathogen in influencing the evolution of ILD associated with emphysema [36]. *Pseudomonas aeruginosa* was demonstrated to have a high prevalence in severe chronic obstructive pulmonary disease (COPD) patients. The main risk factor for the recovery of this microorganism was the recurrent use of antibiotics and the presence of lung bronchiectasis [37]. Moreover, the presence of *Pseudomonas* infection or colonization was often associated with immunosuppressive treatments in patients without bronchiectasis [38]. Usually, *Pseudomonas* colonization is associated with disease severity in patients with primary ciliary dyskinesia [39], chronic bronchiectasis [40,41], and COPD [42,43,44]. *Haemophilus influenzae* is a common bacterium detected in chronic respiratory disease. It possesses a number of different mechanisms to persist into the respiratory system [45]. In COPD eradication or the prevention of *Haemophilus*, infection may lead to a better outcome [46]. Furthermore, the role of this microbe infection and the interaction between the host and the environment has been suggested, as well as its importance in the development, progression, and exacerbation of ILD [13]. Although very few data have been reported on the infection and colonization of NTM organisms and ILD, the prevalence rate of this infection in idiopathic pulmonary fibrosis (IPF) has been found to be higher than in the general population [47]. Increasing susceptibility to atypical mycobacteria infection was reported and associated with a damaged respiratory epithelium and the clearance and persistence of airway inflammation [48,49]. Unfortunately, our data cannot allow us to clarify if a specific CTD or radiologic pattern may favor this kind of infection. However, bacterial colonization must be considered as a dynamic phenomenon. Changes in bacterial load and germ species may be generally present in chronic disease. These phenomena are usually associated with a more rapid decline in respiratory function, inflammation of the airways, and greater severity of the disease [41,50]. Critically, the persistence of colonization by pathogens may lead to chronic inflammation that, in turn, correlates with the severity and progression of the disease [39]. However, the observation that almost a quarter of CTD-ILD patients have been found positive for different bacteria, and that these patients present a worse lung function independently of the origin of bacteria, means that bacterial infections may heavily influence the course of CTD-ILD. This is relevant, considering that these patients should be treated with immunosuppressive therapy, which may favor lung infections. In fact, in a recent study [51] the authors reported that among 322 CTD-ILD patients, 42 died during hospitalization, 17 (40%) of whom died due to lung infection. Risk factors for death during hospitalization were age ≥65 years and the risk of pulmonary infection as a consequence of immunosuppressive therapy, thus making a careful risk/benefit analysis of the types of immunosuppressive therapy to be adopted in these patients necessary.

This is a retrospective study, the weakness of which may be linked to its own nature of observational analysis, even though the results of the E-value provide support to the reliability of our results, as well as to the poor information on comorbidities, treatment, and follow-up. The strength consists in having identified for the first time, to the best of our knowledge, a significant association between the presence of lung bacterial colonization and impairment of lung function tests using diagnostic tools in a real-life clinical context, thus underlining the relevance of lung infections in ILD-CTD for its clinical course and prognosis. 

## 5. Conclusions

In conclusion, lung colonization, irrespective of its origin, plays a relevant role in the severity and mortality of CTD-ILD. The immunosuppressive therapy should be used carefully, paying attention to choosing the least pro-infective treatment [51,52]. Finally, the study of respiratory microbial flora could represent, in patients with CTD-ILD, a prognostic marker which, in turn, could direct more focused intervention and therapeutic actions able to modify the prognosis.

## Figures and Tables

**Figure 1 microorganisms-09-00932-f001:**
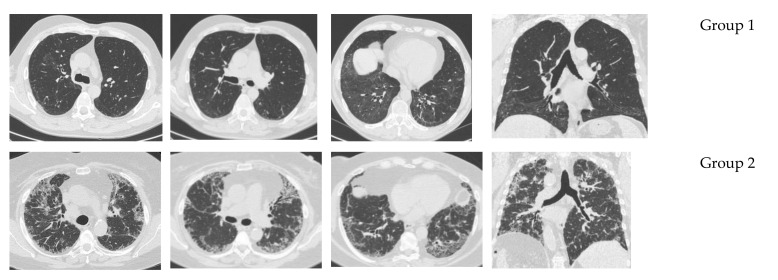
Representative CT scan imaging demonstrating the fibrotic lung involvement in patients belonging to **Group 1** (representing patients in absence of true pathogen in broncho-alveolar lavage) or **Group 2** (representing patients in the presence of pathogens in broncho-alveolar lavage). Note the different fibrotic involvement. Note the ground glass and reticular opacities. Moreover, a relationship between traction bronchiectasis and bacterial colonization was not observed.

**Figure 2 microorganisms-09-00932-f002:**
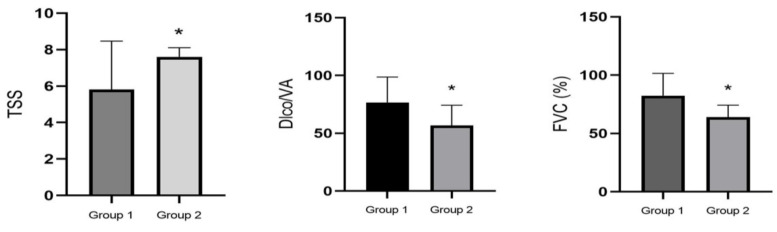
Pulmonary function test findings in patients with CTD-ILD. Group 1 (patients without bacterial colonization in broncho-alveolar lavage) or Group 2 (patients with bacterial colonization in broncho-alveolar lavage). * *p* < 0.05.

**Table 1 microorganisms-09-00932-t001:** Socio-demographic, lifestyle, and clinical characteristics of 44 CTD-ILD patients, examined in univariate analysis.

Characteristic	All N (%)	Infection N (%)	No Infection N (%)	*p* Values
**Sex**				
Males Females	10 (23) 34 (77)	6 (60) 4 (12)	4 (40) 30 (88)	0.005599
**Mean age ± SD**	52.7 ± 16	58 ± 11	47 ± 9	0.0025956
**Smoking** (Mean ± SD P/Y 15 ± 7)				
Current Former Never	24 (55) 14 (32) 6 (13)	4 (17) 4 (29) 2 (33)	20 (83) 10 (71) 4 (67)	NS
**Connective Tissue Disease**				
SSc RA MCTD Overlap syndrome ANCA-vasculitides SS	18 (41) 10 (23) 6 (14) 6 (14) 3 (7) 1	2 (11) 4 (40) 4 (67)	16 (89) 6 (60) 2 (33)	NS vs. SSc 0.00295 vs. SSc
**Comorbidities**				
Diabetes COPD Kidney disease	6 6 4	2 3 0	4 3 4	NS NS NS
**Immunosuppressive drugs**				
Steroids	9	3	6	NS
**Lung HRCT and function**				
TSS Mean FVC % Mean DLCO %		5.8 ± 1.9 67 ± 5 56 ± 14	7.6 ± 2.3 78 ± 20 75 ± 25	0.0398 0.001142 0.008410

CTD-ILD = Connective Tissue Disease-Interstitial Lung Disease; SSc = Systemic Sclerosis; RA = Rheumatoid Arthritis; ANCA = anti-neutrophil cytoplasm antibodies; SS = Sjögren Syndrome; COPD = Chronic Obstructive Pulmonary Disease; FVC = Forced Vital Capacity; DLCO = CO Lung Diffusion; NS = not significant.

**Table 2 microorganisms-09-00932-t002:** Microorganisms found in the broncho-alveolar lavage fluids (>5 mL) of the patients studied.

Pathogens	SSc	RA	MCTD	
**Gram-negative bacteria**				CFU/mL
*Pseudomonas aeruginosa* (*n* = 4)	2		1	10^4^
*Haemophilus influenzae* (*n* = 3)	1	1	2	10^4^
**Slow-growing nontuberculous mycobacteria**				
*Mycobacterium intracellulare* (*n* = 2)	1	1		Polymerase chain reaction- analysis
*Mycobacterium avium* (*n* = 1)		1		

SSc = Systemic Sclerosis; RA = Rheumatoid Arthritis; MCTD = Mixed Connective Tissue Disease; CFU = Colony Forming Units.

**Table 3 microorganisms-09-00932-t003:** Logistic regression multivariate analysis.

Covariates	Lung Infections (Regression CoeffiCient 95% C.I.)	Lung Infections (OR 95% C.I.)	*p*	E-Value
**Male sex**	1.81944 (−0.37/4.25)	6.1684 (0.68/70.2)	0.1	
**DLCO**	−0.1784 (−0.32/−0.07)		0.0035	1.41

DLCO = CO Lung Diffusion; OR = Odds Ratio; C.I. = Confidence Interval.

**Table 4 microorganisms-09-00932-t004:** Logistic regression multivariate analysis.

Covariates	Lung Infections (Regression CoeffiCient 95% C.I.)	Lung Infections (OR 95% C.I.)	*p*	E-Value
**Male sex**	0.9427 (−1.84/3.26)	2.5669 (0.158/47.63)	0.497	
**FVC**	−0.4981 (−0.99/−0.23)		0.0057	1.86

FVC = Forced Vital Capacity; OR = Odds Ratio; C.I. = Confidence Interval.

## Data Availability

Internal database Respiratory Disease Unit Sant’Andrea Hospital Rome, Italy.

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
