# Peer review of "The Role of Lung Colonization in Connective Tissue Disease-Associated Interstitial Lung Disease"

_microorganisms, 2021, doi:10.3390/microorganisms9050932_

Round 1
Reviewer 1 Report
Authors investigated the prevalence of lung bacterial colonization in patients with stable CTD-ILD.
They found that 23% of patients with CTD-ILD had microbial colonization by Pseudomonas aeruginosa, Haemophilus influenzae and non-tuberculous mycobacteria. Authors found that patients with microbial colonization had more severe radiologic lung involvement and higher impairment of respiratory lung function tests compared to the patients without colonization. In their conclusions Authors suggest the possible relationship between bacterial colonization and lung disease progression and severity.
Major criticisms
-The study design is retrospective, so it is not possible to know whether bacterial colonization is a determinant of disease progression and severity or just a complication of a lung with more severe involvement.
-Authors should use the term colonization instead of infection, as their patients did not report any signs of infection (i.e. fever, phlegm, increased dyspnea) at time of BAL.
Minor points
-Pag 2 lines 50-51 in case of acute ILD consider the use of broad-spectrum antibodies, should be changed in broad-spectrum antibiotics.
-Did the Authors observe a relationship between traction bronchiectasis and colonization ?
Author Response
Dear Editor,
We have greatly appreciated the reviewers’ suggestions that have allowed us to significantly improve the revised version of the manuscript. The answers to the criticisms, raised by the reviewers, were detailed below. All the revised parts have been reported in red letters and figures throughout the manuscript.
Reviewer 1
Major criticisms
-The study design is retrospective, so it is not possible to know whether bacterial colonization is a determinant of disease progression and severity or just a complication of a lung with more severe involvement.
The reviewer is right and this point has been discussed and even reported among the weaknesses of the manuscript.
-Authors should use the term colonization instead of infection, as their patients did not report any signs of infection (i.e. fever, phlegm, increased dyspnea) at time of BAL.
We have modified the manuscript; accordingly, the term “infection” was replaced by the term “colonization” in the title and throughout the manuscript when necessary.
Minor points
-Pag 2 lines 50-51 in case of acute ILD consider the use of broad-spectrum antibodies, should be changed in broad-spectrum antibiotics.
The wrong word page 2 line 50-51 was corrected, by replacing “antibodies” with “antibiotics”.
-Did the Authors observe a relationship between traction bronchiectasis and colonization?
No relationship between traction bronchiectasis and colonization was detected. This concept was pointed out in the legend of the Figure 1.
Reviewer 2
Ricci and colleagues present a really nice observational study linking lung infection with worst respiratory function in patients with CTD-ILD. In general the paper is well written and the conclusions are supported by the results. There are however a number of issues that the authors have to address before considering this manuscript for publication.
Since this is an observational study, the major limitation is the influence of unmeasured confounders that could explain away the reported associations. Although the authors clearly state that they could not access to the full medical history to evaluate confounders, the reported associations are based on the untestable assumption that confounding does not influence them. Authors therefore need to perform a sensitivity analysis to address the effect of confounding in the reported associations.
I have strongly appreciated the comments of Reviewer 2 that significantly improved the quality of our results. A sensitivity analysis was carried out, according to the reviewer’s suggestion. The results were detailed in the Results section.
As you can see the limit of observational studies are related to confounders not defined at the beginning of the study. The E-values obtained allow us to define the robustness of the highlighted effect and the strength of our results. Furthermore, the relatively small E-value leads us to hypothesize that a randomized trial could provide more robust results more than a further better defined observational study.
Minor comments:
Please clarify in the methods section in which aliquot clinical microbiology was performed. It looks like that it was done in aliquots 2-4 but it is not clear.
The aliquots used for clinical microbiology examination were the aliquots 2 and 3. This methodological point was underlined in the Materials and Methods section.
- It would be of interest for the readers including a table with the clinical microbiology findings containing, but not limited to, the following information: species detected, method for detection and cfu.
A table including clinical microbiology findings was added and identified as Table 2;
- Please include results for FVC in table 2.
The results of the multivariate logistic regression analysis for FVC were inserted as Table 4;
- Lines 235-236. Pseudomonas aeruginosa and Haemophilus influenza are not normal constituents of lung microbiota but pathogens. Please correct.
The wrong expression was corrected
- Lines 242-288. Please correct the format of those paragraphs.
The paragraphs format was corrected.

Reviewer 2 Report
Ricci and colleagues present a really nice observational study linking lung infection with worst respiratory function in patients with CTD-ILD. In general the paper is well written and the conclusions are supported by the results. There are however a number of issues that the authors have to address before considering this manuscript for publication.
Since this is an observational study, the major limitation is the influence of unmeasured confounders that could explain away the reported associations. Although the authors clearly state that they could not access to the full medical history to evaluate confounders, the reported associations are based on the untestable assumption that confounding does not influence them. Authors therefore need to perform a sensitivity analysis to address the effect of confounding in the reported associations.
Minor comments:
- Please clarify in the methods section in which aliquot clinical microbiology was performed. It looks like that it was done in aliquots 2-4 but it is not clear.
- It would be of interest for the readers including a table with the clinical microbiology findings containing, but not limited to, the following information: species detected, method for detection and cfu.
- Please include results for FVC in table 2.
- Lines 235-236. Pseudomonas aeruginosa and Haemophilus influenza are not normal constituents of lung microbiota but pathogens. Please correct.
- Lines 242-288. Please correct the format of those paragraphs.
Author Response

(The authors gave the same response as above.)

Round 2
Reviewer 2 Report
The authors have properly addressed my comments.
This manuscript is a resubmission of an earlier submission. The following is a list of the peer review reports and author responses from that submission.